# Episodic construction of the early Andean Cordillera unravelled by zircon petrochronology

José Joaquín Jara [1,2,3✉], Fernando Barra [1,3], Martin Reich [1,3], Mathieu Leisen[1,3], Rurik Romero[1,3] & Diego Morata[1,3]

The subduction of oceanic plates beneath continental lithosphere is responsible for continental growth and recycling of oceanic crust, promoting the formation of Cordilleran arcs. However, the processes that control the evolution of these Cordilleran orogenic belts, particularly during their early stages of formation, have not been fully investigated. Here we use a multi-proxy geochemical approach, based on zircon petrochronology and whole-rock analyses, to assess the early evolution of the Andes, one of the most remarkable continental arcs in the world. Our results show that magmatism in the early Andean Cordillera occurred over a period of ~120 million years with six distinct plutonic episodes between 215 and 94 Ma. Each episode is the result of a complex interplay between mantle, crust, slab and sediment contributions that can be traced using zircon chemistry. Overall, the magmatism evolved in response to changes in the tectonic configuration, from transtensional/extensional conditions (215–145 Ma) to a transtensional regime (138–94 Ma). We conclude that an external (tectonic) forcing model with mantle-derived inputs is responsible for the episodic plutonism in this extensional continental arc. This study highlights the use of zircon petrochronology in assessing the multimillion-year crustal scale evolution of Cordilleran arcs.

[1] Departamento de Geología y Centro de Excelencia en Geotermia de los Andes (CEGA), FCFM, Universidad de Chile, Santiago, Chile. [2] Departamento de Ingeniería de Minería, Pontificia Universidad Católica de Chile, Santiago, Chile. [3] Millennium Nucleus for Metal Tracing Along Subduction, FCFM, Universidad de Chile, Santiago, Chile. ✉email: jjjara@uc.cl

Orogenic systems are essential to comprehend the geological history of the Earth[1,2]. Cordilleran orogenic systems occur where oceanic lithosphere subducts under continental crust producing voluminous igneous rocks of mostly intermediate composition[2,3]. Traditionally, the geochemical and isotopic compositions of igneous rocks coupled with field observations, have been used to determine the tectonic and compositional evolution of these continental magmatic arcs[4–6]. However, in some cases, superimposed metamorphic and metasomatic events[7–11] obscure the primary chemical composition of plutonic and volcanic units, precluding the sole use of whole-rock geochemistry to constrain their origin and evolution.

Zircon is an accessory mineral phase commonly found in intermediate and felsic igneous rocks that is resistant to weathering and hydrothermal alteration[12]. It can be precisely dated and is able to incorporate a number of minor and trace elements, a characteristic that has been increasingly used in petrogenetic studies[13,14]. Zircon petrochronology has been applied to comprehend specific magmatic events[15–17] and more recently to study the fertility of intrusive units[18–20], but its potential to trace the evolution of continental arcs at a regional scale has not been fully evaluated.

The early Andean Cordillera of northern Chile, also known as the Coastal Cordillera, represents an early product of the southern Andes[21], one of the most remarkable modern continental arcs in the world[2,5] (Fig. 1a, b). The Coastal Cordillera comprises an orogenic belt that extends parallel to the Peru–Chile trench for more than 1500 km. It was shaped by subduction during multistage, episodic magmatism over more than 100 million years, beginning in the Late Triassic and extending to earliest Late Cretaceous[21,22]. The structural and geodynamic setting of the Coastal Cordillera during the Mesozoic has been the focus of several studies[23–26]. These works have been fundamental to understand the evolution of the southwestern margin of Gondwana[21–27]. Nevertheless, the chemical characteristics and petrogenesis of the Coastal Cordillera plutonic complexes have been rather understudied[22], and the complex relationship between its multiple magmatic episodes, geotectonic changes and magma sources are poorly understood.

Here we report new U-Pb ages and trace element data from zircon grains from several plutonic complexes located along the Coastal Cordillera between 26° and 30°S (Supplementary Fig. 1, Supplementary Data 1 and Supplementary Data 2), covering a timespan of more than 120 million years of geological history of the Andean orogeny. We use a multi-proxy geochemical approach, based on zircon petrochronology and whole-rock analyses, to determine the contributions of mantle-crust-slab-sediments to the magma source in order to better understand the early evolution of the Andean continental arc.

## Results

**The early Andean Cordillera of northern Chile.** The early Andean Cordillera is a non-collisional orogen built on an active subduction margin since the Late Triassic[22]. The magmatic activity progressively migrated eastward and was associated with back/intra-arc basins[21–27] in a multistage, extensional to a transtensional setting[24,25]. This continental arc developed over a Palaeozoic to early Mesozoic basement and was epitomized by thick (6–10 km) basaltic to andesitic Middle Jurassic volcanic units with minor sedimentary sequences, and by north-south oriented, elongated plutonic complexes[28]. The volcanic sequence has a tholeiitic affinity at the bottom and calc-alkaline at the top, and is characterised by primitive Sr, Nd and Pb isotopic compositions[28–31]. However, these rocks are enriched in large ion lithophile elements (LILE), probably because of variable sediment

contributions[30,32]. This thick volcano-sedimentary sequence was intruded by calc-alkaline plutonic complexes distributed in parallel belts of N-S orientation. These belts have a general trend of eastward-younging ages[22,33,34] (Fig. 1b and Supplementary Fig. 1). Back/intra-arc basaltic to andesitic lavas interbedded with marine units of Late Jurassic to Early Cretaceous ages overlay the Middle Jurassic volcanic rocks at its eastern edge[21,28]. During the latest Early Cretaceous, an extensional event led to subsidence and the development of marginal basins[35]. Volcanic and sedimentary rocks, representative of this event, unconformably overlay the back-arc/intra-arc units[21]. At the end of the Early Cretaceous, and concurrent with the final breakup of Gondwana and the opening of the Atlantic Ocean, the extensional regime changed to a compressional one[27], which led to the inversion and closure of back/intra-arc basins and forced the migration of the magmatic arc to the east[27,35].

The main structural feature of the Coastal Cordillera is the Atacama Fault System (AFS). The AFS extends from north to south for more than 1000 km and was active intermittently at least since the Middle to Late Jurassic[19] (Supplementary Fig. 1). This structural system comprises vertical ductile shear zones and brittle fault arrangements in concave segments with north-northwest, north and north-northeast faults with normal and sinistral components[23–26,34].

**Episodic, calc-alkaline plutonism in the early Andean Cordillera of northern Chile.** In the past decades, an increasing number of studies have recognized the episodic nature of magmatic arcs[3,36], suggesting mainly two models of formation: (i) an 'internally controlled' model, based on the interplay between tectonic and magmatic processes and inputs from the upper plate, e.g. forearc underthrusting[3] and landward migration of the arc;[37] and (ii) an 'externally driven' model involving processes such as mantle-flow fluctuations or tectonic reconfigurations[38,39]. Episodic magmatism has mostly been studied in compressional continental arcs, with only a few cases reported in oceanic or continental arcs developed under extensional to transtensional regimes[36]. The available geochronological data from intrusive units in the study area (Supplementary Data 3) supports the notion that these rocks were generated by several magmatic pulses during the eastward migration of the arc[21,27] (Fig. 1b, c). Previous studies[22,26,33,34] identified five possible plutonic episodes between 200 and 95 Ma, with an apparent lull between 180 and 165 Ma and a significant flare-up from 140 to 120 Ma.

These episodes are here constrained using zircon petrochronology on representative plutonic complexes. Zircon dates can be used to identify temporal patterns in long-lived magmatic arcs. A representative sampling of all igneous units coupled with a sound knowledge of the geological record of the study area is desirable to obtain reliable age populations[36]. However, continental orogenic systems such as the Andes are characterised by abundant intermediate to felsic igneous rocks and where mafic compositions are usually underrepresented. In addition, the common presence of zircon xenocrysts yielding highly variable age populations in volcanic rocks makes sampling of plutonic units preferable. Hence, this sampling bias is inevitable, but does not affect the identification of general temporal trends[40].

Our results reveal younging crystallisation ages to the east (Fig. 1b and Supplementary Fig. 1), and clearly define six plutonic episodes at ca. 215–203, 200–185, 160–145, 138–121, 120–108 and 103–94 Ma (Fig. 1d). The differences between outcrop and zircon ages (Fig. 1c, d) can be explained by: (i) the higher number of igneous units and samples included in the outcrop series, which allows better identification of minor events and better assessments of the age amplitude of major ones (sampling bias[36]);

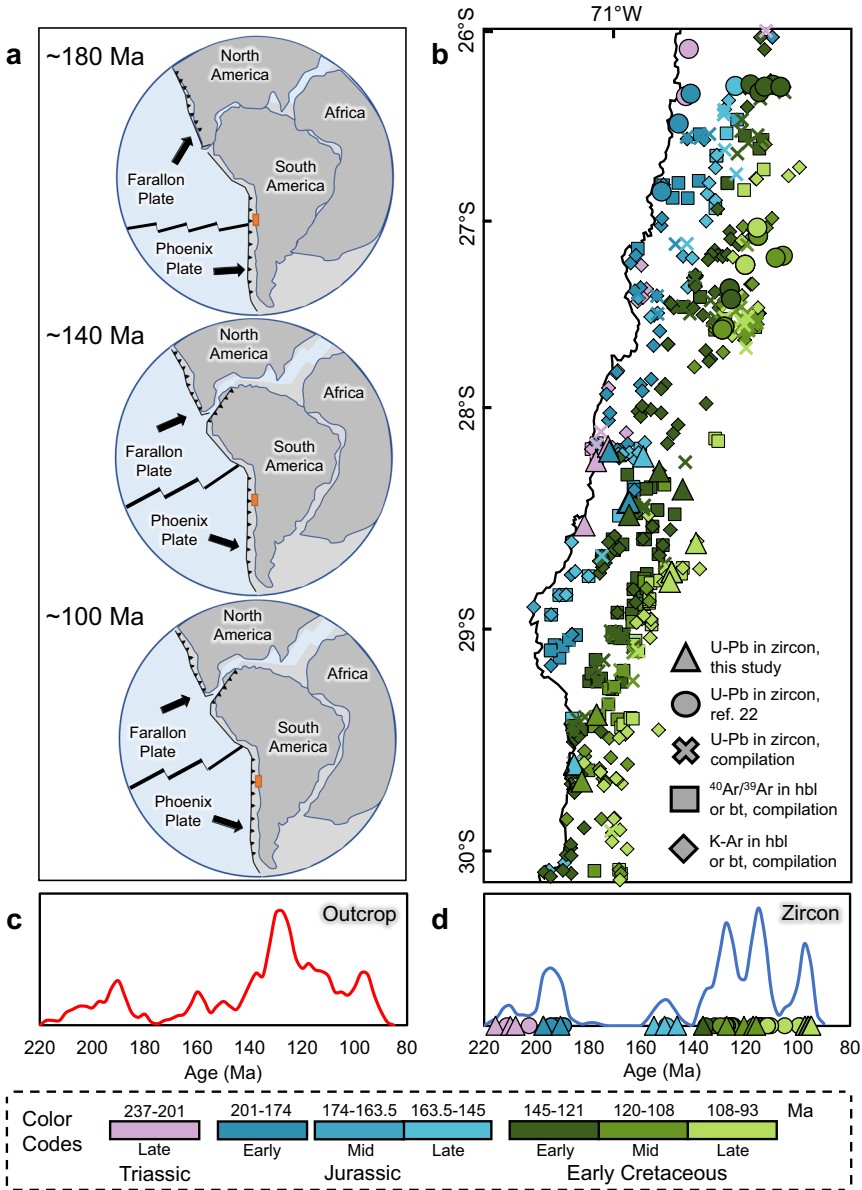

**Fig. 1 Geotectonic setting of the early Andes in northern Chile and outcrop/zircon ages in the study area. a** Geotectonic setting of Gondwana's southwestern margin during the early Andean tectonic cycle (~200–100 Ma). **b** Outcrop ages of plutonic complexes in the study area; symbol colours refer to age of plutonic complexes. **c** Probability distribution function of outcrop ages in the study area. **b, c** Include compiled data reported in the Supplementary Data 1, Supplementary Data 2 and Supplementary Data 3. **d** Probability distribution function of zircon $^{238}$U/$^{206}$Pb ages in the study area, including data from Supplementary Data 1 and Supplementary Data 2. **c, d** Probability distribution functions were calculated using a kernel density estimation process with a bandwidth of 2.5 Ma and a bin width of 5 Ma.

and (ii) the inclusion of crystallisation and cooling ages (K-Ar and $^{40}$Ar/$^{39}$Ar data) in the outcrop series, making it less prone to precisely define the time span of each plutonic episode. Nevertheless, these ages apparently have a minor effect in the analysis due to rapid cooling of plutonic units in the early Andean continental arc[22,34].

The plutonic complexes of the early Andean Cordillera are mainly composed of amphibole and biotite diorites to granodiorites (Supplementary Data 4), with the presence of granites mainly during the inception of the magmatic arc[28,31] (Supplementary Fig. 2a). These complexes are predominantly metaluminous to slightly peraluminous with calc-alkaline affinities (Supplementary Fig. 2b–d). All these rocks are enriched in LILE and show variable fractionation of rare earth elements (REE). Lower Nb, Ta, P and Ti contents are also observed, as well as positive Pb-

anomalies[22,30,31]. These results, coupled to published isotopic data[29,31], suggest that they were generated from a juvenile and depleted mantle source in a subduction setting. Nevertheless, each plutonic episode presents distinct geochemical features that can be used to trace the evolution of the arc[36].

**Zircon and whole-rock petrogenetic indicators**. To better constrain the episodic magmatism and trace the evolution of the early Andean continental arc, we selected 10 petrogenetic indicators based on whole-rock data (Fig. 2) and zircon chemistry (Fig. 3). First-order tectonic parameters were determined by: (i) whole-rock La$_N$/Yb$_N$ (Fig. 2a) and Sr/Y ratios (Supplementary Fig. 3), which are used to estimate crustal thickness or depth to Moho;[41,42] (ii) zircon Th/U ratios (Fig. 3a), to identify extensional and compressional periods in subduction environments;[43] and

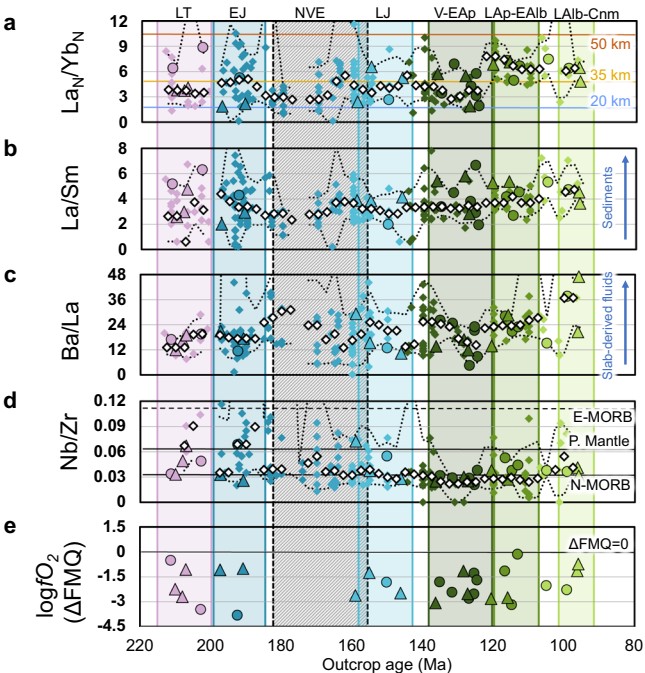

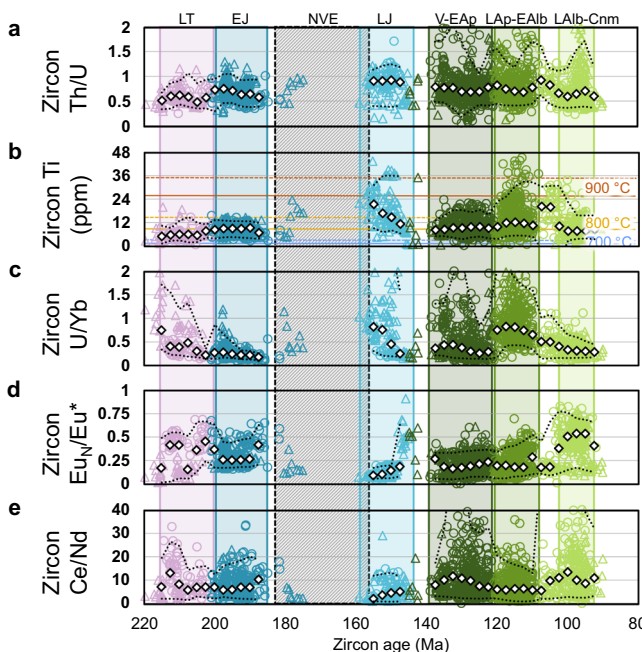

**Fig. 2 Whole-rock petrogenetic indicators.** The geochemical data are used to estimate the following parameters: **a** crustal thickness[41]. **b**, **c** Sediment and slab-derived fluid contributions to the magma source[4]. **d** Enrichment of the magmatic source[4]. Reference values for N-MORB, Primitive Mantle (P. Mantle) and E-MORB are also shown[60]. **e** Oxygen fugacity (ΔFMQ) was determined by rock-to-zircon partitioning coefficients[46]. All geochemical parameters are plotted according to the outcrop age. Data source: triangles (this study: Supplementary Data 1); circles (ref. [22] Supplementary Data 2), rhomboids (compiled whole-rock analyses: Supplementary Data 3). Symbol colours refer to the age of the plutonic complexes as in Fig. 1. Rhomboidal white symbols and lower and upper dotted black lines represent median, 5th and 95th percentiles of each series (kernel density estimation with 2.5 Ma bandwidth and 5 Ma bin width, except for calculated oxygen fugacity). LT, Late Triassic (215–203 Ma). EJ, Early Jurassic (200–185 Ma). LJ, Late Jurassic (160–145 Ma). V-EAp, Valanginian to early Aptian (138–121 Ma). LAp-EAlb, late Aptian to early Albian (120–108 Ma). LAlb-Cnm, late Albian to Cenomanian (103–94 Ma). Colour bands are according to Fig. 1. NVE, La Negra volcanic event (~180–155 Ma), is represented by the dashed grey band.

**Fig. 3 Zircon petrogenetic indicators.** The trace element data in (**a**, **c–e**) and the zircon Ti content (**b**) are used to estimate the following parameters: **a** extensional and compressional periods in subduction environments[43]. **b** Crystallisation temperatures;[44] reference lines consider $a[SiO_2] = 1$ and $a[TiO_2] = 0.5$ (solid lines) and 0.7 (dashed lines)[45]. **c** Enrichment and/or crustal contamination of the magmatic source[13]. **d** Oxygen fugacity changes and plagioclase fractionation prior to zircon crystallisation. $Eu^* = [Sm_N \times Gd_N]^{0.5}$. **e** Oxygen fugacity during zircon crystallisation. All geochemical parameters are plotted according to their corresponding zircon $^{238}U/^{206}Pb$ age. Data source: triangles (this study: Supplementary Data 1); circles (ref. [22] Supplementary Data 2). Symbol colours refer to the age of the plutonic complexes as shown in Fig. 1. Rhomboidal white symbols and lower and upper dotted black lines represent median, 5th and 95th percentiles of each series (kernel density estimation with 2.5 Ma bandwidth and 5 Ma bin width, except for (**e**). LT, Late Triassic (215–203 Ma). EJ, Early Jurassic (200–185 Ma). LJ, Late Jurassic (160–145 Ma). V-EAp, Valanginian to early Aptian (138–121 Ma). LAp-EAlb, late Aptian to early Albian (120–108 Ma). LAlb-Cnm, late Albian to Cenomanian (103–94 Ma). Colour bands according to Fig. 1. NVE, La Negra volcanic event (~180–155 Ma), is represented by the dashed grey band.

(iii) Ti-in-zircon temperatures (Fig. 3b), which could be linked to magma composition, depth of emplacement and cooling rates in magma chambers[44,45]. In addition, the magma source was determined by whole-rock La/Sm, Ba/La and Nb/Zr[4] and zircon U/Yb[13] elemental ratios (Fig. 2b–d and Fig. 3c, respectively). The redox state of magmas is recorded by zircon Eu-anomalies and Ce/Nd ratios (Fig. 3d, e). It is also calculated by using rock-to-zircon elemental partitioning coefficients[46,47] and expressed in logarithmic units of oxygen fugacity relative to the fayalite–magnetite–quartz mineral buffer (ΔFMQ) (Fig. 2e). To discern trends and/or anomalous values, the median and 5th and 95th percentiles were calculated using a kernel density estimation with a bandwidth of 2.5 Ma and a bin width of 5 Ma for each series, except for the oxygen fugacity due to the limited number of samples with zircon and whole-rock analyses ($n = 30$). In addition, the median and 5th and 95th percentiles for the 10 petrogenetic indicators and by each plutonic episode are reported in Supplementary Table 3.

**Multistage evolution of the early Andean Cordillera.** The construction of the early Andean Cordillera commenced in the

Late Triassic, when subduction restarted along the southwestern margin of Gondwana after an anorogenic period[21,22]. The magmatic arc was established in the current Coastal Cordillera in a rather attenuated crust[31] within multistage, extensional to transtensional tectonic settings[24,25], which led to six plutonic episodes at 215–203, 200–185, 160–145, 138–121, 120–108 and 103–94 Ma (Fig. 1c, d). These episodes are characterised using the proposed indicators (Figs. 2 and 3) and schematically represented in cross sections of the continental arc in Fig. 4.

The first and second plutonic episodes of the early Andean continental arc took place between 215–203 and 200–185 Ma (LT and EJ; Fig. 1c, d, and Figs. 2 and 3). These episodes are marked by a gradual and continuous thinning of the continental crust from ~30 to ~22 km (Fig. 2a and Supplementary Fig. 3) and a depletion of the magmatic source. Zircon grains display low Th/U ratios with minor dispersion (Fig. 3a), a behaviour usually linked to convergent settings[43]. Ti-in-zircon temperatures vary within a restricted range (Fig. 3b). Minor sediment contributions and crustal contamination are recorded by the La/Sm ratio (Fig. 2b) and fluid-derived elements from slab dehydration are negligible (Fig. 2c). This led to an increasingly depleted source with a

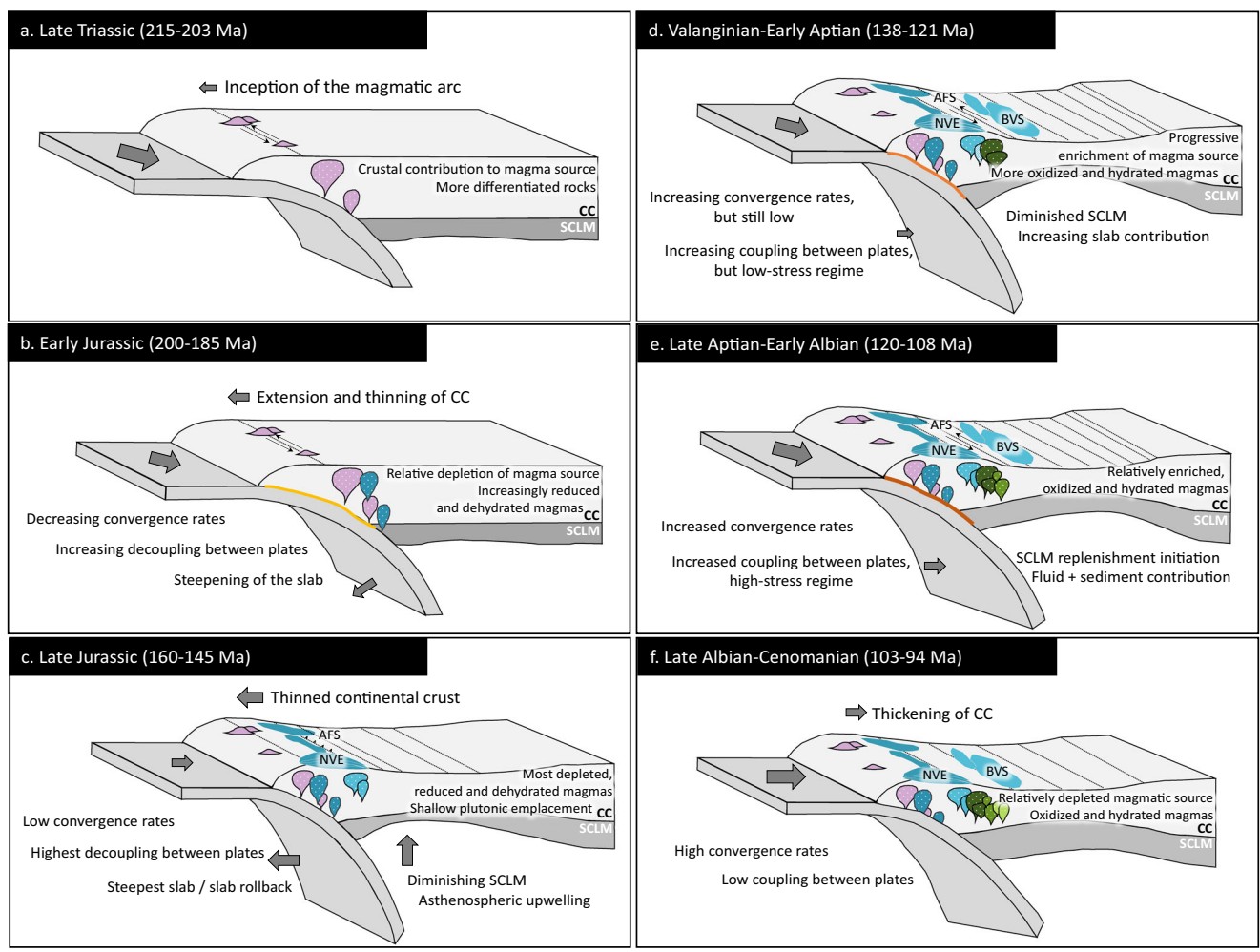

**Fig. 4 Multistage evolution of the early Andean Cordillera. a** Late Triassic plutonic episode (215–203 Ma): represents the inception of the early Andean magmatic arc in an attenuated continental arc, with crustal inputs to the magma source and the emplacement of granitic intrusions. **b, c** Early Jurassic to Late Jurassic plutonic episodes (200–145 Ma): this transtensional to extensional period implied a decrease in sediment, slab-derived fluids and lithospheric mantle contributions to the magma source, decompressional melting and upwelling of asthenospheric material. This led to a progressively depleted, dehydrated and reduced magmatism during the Late Jurassic. **d, e** Valanginian to late Aptian plutonic episodes (138–108 Ma) showed the transition from an arc-normal extensional to a transtensional setting with probably higher convergence rates and increasing plate coupling. As a result, subduction contributions increased steadily over time, generating increasingly more enriched, hydrated and oxidized magmas. **f** Late Albian to Cenomanian plutonic episode (103–94 Ma) derived from moderately oxidized magmas and a relatively depleted mantle source related to high convergence rates and plate decoupling. CC, continental crust. SCLM, sub-continental lithospheric mantle. AFS, Atacama Fault System. NVE, La Negra volcanic event. BVS, back-arc volcano-sedimentary and sedimentary sequences.

N-MORB signature that is reflected in the Nb/Zr (Fig. 2d) and zircon U/Yb (Fig. 3c) ratios. The redox state of the magmas was relatively low due to the diminishing effect of slab fluids. A slight decrease in oxygen fugacity values is observed during this period (Figs. 2e and 3d, e). These geochemical signatures suggest that the emplacement of the plutonic complexes took place in an attenuated margin[21,27] during a transtensional regime with an increasing subduction angle and plate decoupling[24], which led to a gradual but persistent thinning of the continental crust (Fig. 4a, b). These results are consistent with structural observations in the forearc[24], and an isotopic 'pull-up' of its igneous units[29,31] which is uncommon in continental arcs[48]. This period ended with foundering of the slab, subduction rollback and a complete decoupling of plates in an arc-normal extensional regime during the extensive La Negra volcanic event, NVE (~180 to ~155 Ma; grey band in Figs. 2 and 3)[24,27].

The third episode (160–145 Ma; LJ; Fig. 1c, d, and Figs. 2 and 3) partially overlaps with La Negra volcanic event and occurs prior to a plutonic hiatus in the Jurassic to Cretaceous transition. During this period, a crustal thickness of ~20 km is estimated based on whole-rock chemistry (Fig. 2a and Supplementary Fig. 3) and the highest zircon Th/U ratio among the studied samples (Fig. 3a). High crystallisation temperatures (~850 °C) are also recorded (Fig. 3b). Accordingly, Late Jurassic magmatism would have been generated primarily by decompressional melting of the upper mantle[28,49] with minor contributions from the subduction process (Fig. 2b, c). This is reflected in a highly depleted and reduced signature with whole-rock Nb/Zr values consistently near or below the N-MORB value (Fig. 2d), a median $\Delta$FMQ of −2.15 (Fig. 2e), and zircon $Eu_N/Eu^*$ and Ce/Nd ratios with median values of 0.11 and 3.8, respectively (Fig. 3d, e). Nevertheless, a relatively high zircon U/Yb ratio was obtained due to the influence of two highly differentiated samples, uncommon in this geological period (samples JJJD_43 and JJJD_62). Excluding these samples, a U/Yb ratio of 0.27 was obtained, which is similar to that of the Early Jurassic episode (Fig. 3c). These results are consistent with: (i) a strong arc-normal extensional setting and shallow plutonic emplacement, determined by structural analyses of the AFS and

its relationship with surrounding plutonic bodies;[24,25,34] and (ii) a negative to neutral, trench normal absolute and relative convergence rates in the kinematic plate model of South America[50]. Overall, during the long-lasting, transtensional to extensional transition that occurred between the Late Triassic and Late Jurassic, the contribution of sediments and slab-derived fluids to the magmatic source was strongly diminished. In addition, the lithospheric mantle thickness was reduced triggering decompressional melting and upwelling of asthenospheric material to the base of the crust (Fig. 4c). This led to a progressively depleted, dehydrated and reduced magmatism[22,28–30,49].

The fourth and fifth plutonic episodes occurred between 138–121 and 120–108 Ma, respectively (V-EAp and LAp-EAlb; Fig. 1c, d, and Figs. 2 and 3). These episodes are characterised by increasing crustal thickness, reaching over ~35 km (Fig. 2a and Supplementary Fig. 3), slightly lower crystallisation temperatures (Fig. 3b) and higher fluid and sediment contributions to the mantle wedge compared to the previous period (Fig. 2b, c). As a result, the magmas became more enriched (Fig. 3c), hydrated and oxidized (Fig. 3d, e). These trends could be explained by: (i) the transition from an arc-normal extensional to an oblique (low-stress?) transtensional regime (Fig. 4d), a change that has been recorded in the kinematics of the AFS;[24–26,34] and (ii) increased coupling between plates, likely due to higher convergence rates[24,50] (Fig. 4e). Consequently, subduction contributions (sediments and fluids) progressively increased and magmas became more enriched, hydrated and oxidized than in previous episodes.

The last plutonic episode of the early Andean Cordillera occurred during the late Albian to Cenomanian (103–94 Ma; LAlb-Cnm; Figs. 2 and 3) after a magmatic lull in the middle Albian (108–103 Ma; Fig. 1c, d). An estimated crustal thickness of >35 km (Fig. 2a and Supplementary Fig. 3) coupled with increasing contributions of slab-derived fluids and lesser sediments (Fig. 2b, c) as well as a depleted mantle source (Figs. 2d and 3c), could be better explained by an extensional to transtensional regime with high convergence rates and relative decoupling of plates (Fig. 4f). This interpretation is also supported by relatively high oxygen fugacity markers (Figs. 2e and 3d, e) and low crystallisation temperatures (Fig. 3b). These results are consistent with the geological record of the back-arc[35], and with changes in trench normal and parallel relative convergence rates from the kinematic plate model[50].

## Discussion

The evidence presented in this study, coupled with current knowledge of the tectonic and structural evolution of Gondwana's southwestern margin, allows us to conclude that the episodic magmatism of the early Andean Cordillera resulted from a multistage, transtensional to extensional subduction setting[21,24,28]. Six plutonic episodes were identified between the Late Triassic and earliest Late Cretaceous (215–94 Ma); each one related to specific substages in the evolution of the continental arc. These episodes are better identified and characterised based on zircon petrochronology coupled with whole-rock geochemical analyses (Figs. 2 and 3), and can be associated with significant tectonic changes in the continental margin (Fig. 4). Therefore, an "external forcing" model[36] with mantle-derived inputs[39] is argued as the mechanism for the episodic plutonism in this extensional continental arc.

Our interpretations are in agreement with a proposed Cretaceous (ca. 120–90 Ma) flare-up event recorded in the Western Peninsular Ranges Batholith (US), the Peruvian Coastal Batholith, and the Costal Cordillera of central Chile south of the study area[39]; and with the episodic magmatism in the Mesozoic Median Batholith of Fiordland, New Zealand[51]. Further, the new data presented here are consistent with intermittent magmatism in the American Cordilleras[52]. However, they contrast with data reported for other continental arcs developed in compressional to transpressional tectonic settings, such as the Cenozoic central Andes[53], the North[37,54–56] and South[57] American Cordilleras, and the collisional Gangdese Batholith[58]. In those, an 'internal feedback' model[36], based on processes such as foreland underthrusting[3] or arc migration[37], is the preferred explanation for cyclical magmatism. These dissimilarities could be attributed to different tectonic processes governing the arcs during flare-ups[38,57], and could explain the variability among high flux events in a particular magmatic arc through time, as in the case of the Mexican Cordillera[59]. Consequently, episodic magmatism in continental arcs could be explained by: (i) external forces (first-order variables) without a significant modulating effect of crustal-scale processes (second-order variables) during an extensional tectonic regime; and (ii) a variable mixture of external forces and intra-arc feedbacks in compressional settings[38,52].

Our results highlight the use of zircon petrochronology as a potential tool to unravel the multimillion-year crustal scale evolution of Cordilleran arcs.

## Methods

**Sampling and analytical methods**. Descriptions of the studied plutonic complexes and analysed samples are presented in Supplementary Table 1 and Supplementary Table 2, respectively. Methods for whole-rock analysis and for simultaneous zircon U-Pb geochronology and trace element determinations are reported in the Supplementary Material. Raw data of the whole-rock and zircon analysis from this study are shown in Supplementary Data 1, and for ref. [22] in Supplementary Data 2. Compiled published radiometric ages for plutonic units in the study area are in Supplementary Data 3. Compiled published whole-rock analyses for intrusive rocks in the early Andean Cordillera are in Supplementary Data 4.

## Data availability

The authors declare that all relevant data are available within the article and its Supplementary Information files.

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

## Acknowledgements

Funding for this study was provided by FONDECYT grant #1190105 to F.B and M.R. Additional support was provided by ANID through Millennium Science Initiative Program (NCN13_065) and FONDAP project #15090013 'Centro de Excelencia en Geotermia de Los Andes, CEGA'. José Joaquín Jara thanks ANID for providing support through a Ph.D. scholarship. The analytical work was supported by FONDEQUIP instrumentation grant EQM120098. We thank Dr. José Cembrano for comments on an early version of this manuscript and Dr. John Browning who helped with final copy editing.

## Author contributions

J.J.J.: Conceptualization, methodology, formal analysis, investigation, resources, writing—original draft, writing—review and editing, visualization. F.B.: Conceptualization, methodology, validation, investigation, resources, writing—review and editing, supervision, project administration, funding acquisition. M.R.: Validation, investigation, resources, writing—review and editing, supervision, funding acquisition. D.M.: Methodology, validation, writing—review and editing, supervision, funding acquisition. M.L.: Software, validation, formal analysis, investigation, resources, writing—review and editing. R.R.: Software, validation, formal analysis, investigation, resources, writing—review and editing.

## Competing interests

The authors declare no competing interests.

**Additional information**

