## [Peer Review File · Nature Communications]

REVIEWERS' COMMENTS

Reviewer #1 (Remarks to the Author):

Specific line comments:

Line 19-21: The subduction of oceanic plates beneath continental lithosphere resulting in Cordilleran volcanic arcs is responsible for.....

Line 22: Need a sentence joining the first sentence to the "Here we use...." Perhaps introducing the Andes.

Line 23: "the Andes, the most remarkable modern continental arc" is a subjective statement.

Line 26: list out the approx ages of the plutonic episodes. Might help someone surfing abstracts find what they are looking for.

Line 30-31: Could you be more specific about the mechanics responsible for episodic plutonism? Add specifics?

Line 32: Spell out "multi-million year"

line 39-40: subject verb agreement/plural. Compositions....have been.

Line 50-51: this seems like an overstatement. Reword.

Line 53-54: "remarkable modern..." this was used earlier and is subjective.

Line 73-75: The magmatic arc is intrinsically associated with back/intra-arc basin? Explain? Just a typical relationship between an arc and back arc?

Line 76: "thick" how thick?

Line 92: Is Atacama fault system a proper name?

Line 137: include Sr/Y in the list, no parentheses.

Line 135: 10 petrogenic indicators: only three listed as first order. There appear to be 6 more in the "In addition" list. Perhaps make this more clear.

There is very little discussion of the data trends before jumping into the discussion.

Methods are mentioned at the end and there is no conclusion. A clearly marked Conclusion/Summary paragraph would help significantly.

Unless Nature Communications has a different format, this is not the traditional flow of manuscripts, which will confuse some readers.

The flow is very jumpy and jumps right into the interpretation without any real talk of the data, then ends rapidly.

It's pretty difficult to assess the interpretation, because it jumps right into presenting the data and interpreting the magmatic pulses. It might be easier to read if the trends of each petrogenic indicator are discussed first through time, THEN woven into a story about the evolution of the magma system. The Figure 2 elements are too small to really assess. Might increase size and add a few y axis lines to help the reader follow along.

The science is there, this just needs some work on flow and a slight rearranging of topics.

Reviewer #2 (Remarks to the Author):

I found this paper to be extremely informative, useful for my own research, and I would claim for anybody working in the Andes or with magmatic arcs. There are lots of new data nicely presented and put to work towards a bigger goal: unravel the early Andean evolution which is preserved today in the coastal cordillera. I am also extremely fond of a group that promotes exceptional research even though it does not come from the richest parts of the globe which in itself is remarkable and worthy of mentioning.

Writing is good, not great, can be improved. I think that is the biggest issues with the manuscript but given the short deadline the journal gave me to work with, I cannot possibly do a thorough reading and combing of small things. The authors can do that themselves or give the manuscript to somebody to fix it that way. It's not a big deal.

Other than that, some comments on why zircon abundance equals magmatic fluxes would be useful. Some of the more mafic parts of arcs are not as fertile in zircons, etc.

This is an excellent manuscript which can be published after some minor changes. I enjoyed reading it and look forward to seeing it published. I would cite this paper a lot. We have an ongoing central Andean NSF project funded (a large multi disciplinary and multi annual type project) and what is in this paper is golden for our discussions, our hypotheses and our approaches in general.

Respectfully, Mihai Ducea , Tucson AZ

Reviewer #3 (Remarks to the Author):

Review of Jara et al. Nature communications.

This paper presents a new and relatively large dataset of geochemistry and geochronology from Early Andean samples from Northern Chile. This is a well formulated and interesting study suggesting pulsed magmatic activity in this region of the Andes which supports the idea that the rocks were formed due to the interplay of tectonic and magmatic processes. The samples are broken apart by age, and each age peak are thought to represent a different tectonic event, each producing magmatism with different chemistry and also zircon crystals with different compositions. The evolution of the Andes is a subject that has been discussed a great deal in the literature with multiple different models proposed for its formation. Here Jara et al. place their data in the tectonic context outlined in previous studies, and show that the data align best with an external forcing model for the formation of the units. The data are well presented, with nice figures and the text is relatively well written with a few minor English mistakes. I'm not a specialist on Andean geology, and therefore cannot comment on the specifics of the model, however the model appears to build on the similar dataset previously published by the authors (Jara et al. 2021 doi.org/10.1016/j.gr.2021.01.007). The previous dataset was collected from samples in the northern half of the study region, this current study presents new data from the south and combines it with the previously published data from the north and south of the region. The main issue with the paper is that the new dataset presented here don't really bring a new model forward for the formation of these rocks, or a new tectonic interpretation, the data agree with previous models. Normally this wouldn't be an issue at all for a manuscript, but when it is submitted to Nature communications, the paper should bring a fundamental change to our understanding of the area, and it should have wide reaching implications. I'm not sure that this is the case here. There is no link between the tectonic model for this area and that for the rest of the Andes, or for this type of subduction zone setting elsewhere in the world – both of these would significantly increase the impact of the current study.

Some smaller comments -

The zircon data appear to fit with the WR geochemical data, although I'm relatively sure that the same conclusions could be drawn from the WR data themselves without the need for the zircon chemistry data,

apart from the oxygen fugacity data. The oxygen fugacity data presented are also not necessarily in full agreement with the statements in the manuscript – for example, the on line 171, the authors indicate the the oxygen fugacity decreases during the first two magmatic episodes, however the values fluctuate between ~ -1.5 to ~ -4 , the zircon Ce anomaly doesn't change much nor does the Eu anomaly, which has recently been linked to depth of melting, so may not only be effected by oxygen fugacity (Tang et al. 2020 - doi.org/10.1130/G47745.1).

The authors suggest that the zircon trace element data is required because there is the possibility that the WR data has been altered, however the WR data are the data that most clearly show the origin of the samples and the different conditions in the source. The WR data could be also used to constrain the Ti activity, for the Ti temperature calculations. The Ti activity is currently treated in a relatively simplistic manner – I assume the same value is used for all of the samples? Is this assumption justified? Could changes in rock type, mineralogy, water content, play a role here?

The difference between the age distribution for the WR samples and those for the zircon grains could be explained by problems with the K-Ar data? These could be removed. Are all of the zircon ages plotted in the zircon distribution diagram or just the zircons thought to date the emplacement of the intrusion? Its not clear from the figures presented here, but are there antecrystic zircon or xenocrystic zircon, or zircon with suspected Pb loss present in the datasets? Are these plotted too? Are these data also included in the trace element diagrams? Are cut offs used for zircon trace element concentrations that could be affected by secondary alteration (see Bell et al. 2019 - doi.org/10.1016/j.chemgeo.2019.02.027)?

I'm not sure why the data from Jara et al. 2021 are included in the zircon data table here? these should be included in the data table with the previously published data.

The secondary reference data need to be included for trace element and U-Pb data.

Reviewers' comments:

Dear Dr Jara,

Your manuscript entitled "Episodic construction of the early Andean Cordillera unraveled by zircon petrochronology" has now been seen by our referees, whose comments appear below. In light of their advice I am delighted to say that we are happy, in principle, to publish a suitably revised version in Nature Communications under the open access CC BY license (Creative Commons Attribution 4.0 International License).

Congratulations on this achievement, this must be the one paper per year which I can accept after one round of review! The reviewers still do have a number of comments which I would ask you to work on. Most importantly, they ask you to overhaul the flow and structure of your manuscript to improve its readability. Also, please make use of a professional English proofreading service. Towards referee #3, please address in particular oxygen fugacity data, Ti activity assumptions, treated simplistic plus problems between the age distribution for the WR samples and those for the zircon grains. Remember not to only address all points raised in a rebuttal letter, but to also implement them in the main text.

We therefore invite you to revise your paper one last time to address the remaining concerns of our reviewers and our editorial requests in the attached document(s). At the same time we ask that you edit your manuscript to comply with our policies and formatting requirements and to maximise the accessibility and therefore the impact of your work.

Please see the attached document(s), listing a number of points that must be addressed. Failure to comply with our editorial requests will cause delays in accepting your manuscript. Please also see the Nature Communications formatting instructions for further information.

REVIEWERS' COMMENTS

Reviewer #1 (Remarks to the Author):

Specific line comments: extensive

Line 19-21: The subduction of oceanic plates beneath continental lithosphere resulting in Cordilleran volcanic arcs is responsible for.....

A: Changed according to the reviewer's suggestion.

Line 22: Need a sentence joining the first sentence to the "Here we use...." Perhaps introducing the Andes.

A: Agreed. We have added a sentence according to the reviewer's suggestion.

Line 23: "the Andes, the most remarkable modern continental arc" is a subjective statement.

A: We have modified the sentence to "... the Andes, one of the most remarkable modern continental arcs of the world".

Line 26: list out the approx ages of the plutonic episodes. Might help someone surfing abstracts find what they are looking for.

A: We agree that this information could be useful. However, we would need to expand the abstract far beyond the length limits suggested by Nature Communications.

Line 30-31: Could you be more specific about the mechanics responsible for episodic plutonism? Add specifics?

A: Details of the mechanism resulting in episodic magmatism are provided in the Results section, but we did not include them in the abstract due to length limitations.

Line 32: Spell out "multi-million year"

A: Changed according to the reviewer's suggestion.

line 39-40: subject verb agreement/plural. Compositions....have been.

A: Corrected.

Line 50-51: this seems like an overstatement. Reword.

A: We have changed the word "extensively" by "recently".

Line 53-54: "remarkable modern..." this was used earlier and is subjective.

A: Several articles (e.g. Haschke et al. 2006; Ramos 2009; Ducea et al. 2015) refer to the Andes as the most remarkable modern continental arc due to its extension and long-lasting magmatic activity.

Line 73-75: The magmatic arc is intrinsically associated with back/intra-arc basin? Explain? Just a typical relationship between an arc and back arc?

A: We have modified the sentence to clarify the relation between arc and back/intra arc. Basically, we have deleted the word "intrinsically".

Line 76: "thick" how thick?

A: About 6-10 km. This range of thickness values has been included in the text.

Line 92: Is Atacama fault system a proper name?

A: Yes, this is the proper name of this important structural system. See for example articles 23 and 26 in the reference list.

Line 137: include Sr/Y in the list, no parentheses.

A: Agreed. However, the Sr/Y ratio is analogous to the La_N/Yb_N ratio and hence, it is not incorporated in Figure 2, but as a figure in the Supplementary file (Figure 3).

Line 135: 10 petrogenic indicators: only three listed as first order. There appear to be 6 more in the "In addition" list. Perhaps make this more clear.

A: Agreed. Of the 10 petrogenetic indicators, three are first order tectonic parameters. The word "tectonic" was missing in the original manuscript and was added in this version. We have slightly modified the sentence to make it clear.

There is very little discussion of the data trends before jumping into the discussion. Methods are mentioned at the end and there is no conclusion. A clearly marked Conclusion/Summary paragraph would help significantly.

Unless Nature Communications has a different format, this is not the traditional flow of manuscripts, which will confuse some readers.

The flow is very jumpy and jumps right into the interpretation without any real talk of the data, then ends rapidly.

It's pretty difficult to assess the interpretation, because it jumps right into presenting the

data and interpreting the magmatic pulses. It might be easier to read if the trends of each petrogenetic indicator are discussed first through time, THEN woven into a story about the evolution of the magma system.

A: We let the reviewer know that we followed the editorial format for Nature Communications, i.e., introduction, results (subheadings), discussion (optional) and methods (optional, with subheadings and located after Discussion section and before References).

In more detail, and according to Nature Communication guidelines:

"The main text (not including abstract, Methods, References and figure legends) is limited to 5,000 words. ... The main text of an Article should begin with a section headed Introduction of referenced text that expands on the background of the work (some overlap with the abstract is acceptable), followed by sections headed Results, Discussion (if appropriate) and Methods (if appropriate). The Results and Methods sections should be divided by topical subheadings; the Discussion should be succinct and may not contain subheadings. Methods are typically less than 3000 words".

Regardless, we have made our best effort to accommodate the reviewer's suggestion, and we have expanded parts of the manuscript to clarify some aspects of the results and their implications. Moreover, we extended the last subsection to clearly state the conclusions of the study and how it relates to other continental arcs.

The Figure 2 elements are too small to really assess. Might increase size and add a few y axis lines to help the reader follow along.

A: Agreed. In this revised version, we have separated former Figure 2 as two figures to improve its legibility. The new Figure 2 presents the whole-rock petrogenetic indicators and Figure 3 the zircon-based indicators. We have incorporated y axis lines as suggested by the reviewer.

The science is there, this just needs some work on flow and a slight rearranging of topics.

A: We appreciate the positive comments of the reviewer that helped improve the clarity and quality of our contribution.

Reviewer #2 (Remarks to the Author):

I found this paper to be extremely informative, useful for my own research, and I would claim for anybody working in the Andes or with magmatic arcs. There are lots of new data nicely presented and put to work towards a bigger goal: unravel the early Andean evolution which is preserved today in the coastal cordillera. I am also extremely fond of a group that promotes exceptional research even though it does not come from the richest parts of the globe which in itself is remarkable and worthy of mentioning.

Writing is good, not great, can be improved. I think that is the biggest issues with the manuscript but given the short deadline the journal gave me to work with, I cannot possibly do a thorough reading and combing of small things. The authors can do that themselves or give the manuscript to somebody to fix it that way It's not a big deal.

A: The new manuscript was revised by a native English speaker. We hope this effort could have improved the article.

Other than that, some comments on why zircon abundance equals magmatic fluxes would be useful. Some of the more mafic parts of arcs are not as fertile in zircons, etc.
A: This is a very good point. We have added a paragraph to address in more detail our sampling strategy.

This is an excellent manuscript which can be published after some minor changes. I enjoyed reading it and look forward to seeing it published. I would cite this paper a lot. We have an ongoing central Andean NSF project funded (a large multi disciplinary and multi annual type project) and what is in this paper is golden for our discussions, our hypotheses and our approaches in general.

Respectfully, Mihai Ducea , Tucson AZ

A: We sincerely thank the positive comments by Dr. Ducea.

Reviewer #3 (Remarks to the Author):

Review of Jara et al. Nature communications.

This paper presents a new and relatively large dataset of geochemistry and geochronology from Early Andean samples from Northern Chile. This is a well formulated and interesting study suggesting pulsed magmatic activity in this region of the Andes which supports the idea that the rocks were formed due to the interplay of tectonic and magmatic processes. The samples are broken apart by age, and each age peak are thought to represent a different tectonic event, each producing magmatism with different chemistry and also zircon crystals with different compositions. The evolution of the Andes is a subject that has been discussed a great deal in the literature with multiple different models proposed for its formation. Here Jara et al. place their data in the tectonic context outlined in previous studies, and show that the data align best with an external forcing model for the formation of the units. The data are well presented, with nice figures and the text is relatively well written with a few minor English mistakes.

A: We thank the positive comments of this reviewer. This new version was also revised by native English speaker who has a PhD in geology and has expertise in volcanology and rock mechanics.

I'm not a specialist on Andean geology, and therefore cannot comment on the specifics of the model, however the model appears to build on the similar dataset previously published by the authors (Jara et al. 2021 doi.org/10.1016/j.gr.2021.01.007). The previous dataset was collected from samples in the northern half of the study region, this current study presents new data from the south and combines it with the previously published data from the north and south of the region. The main issue with the paper is that the new dataset presented here don't really bring a new model forward for the formation of these rocks, or a new tectonic interpretation, the data agree with previous models. Normally this wouldn't be an issue at all for a manuscript, but when it is submitted to Nature communications, the paper should bring a fundamental change to our understanding of the area, and it should have wide reaching implications. I'm not sure that this is the case here.

A: Several articles have studied the structural systems and the tectonic evolution of the early Andes (Late Triassic to earliest Late Cretaceous) in northern Chile. Nevertheless, the geochemistry of its intrusive rocks and the conditions and characteristics of its magmatism has been rather understudied. Moreover, most geochemical datasets of

the early Andean Cordillera of northern Chile (Late Triassic to earliest Late Cretaceous) lacks precise radiometric ages, precluding a proper study of the evolution of this magmatic arc.

As in all fields of science, this work compiles and uses previously published data, but also presents new zircon U-Pb ages for several intrusive units within this major domain and relates them with whole-rock and zircon trace element determinations. This is probably the first study that relates magmatism geochemistry to changes in the tectonic regime that occur in this remarkable continental arc during more than 120 million years of geological evolution. That is the main contribution of the article.

In addition, we do not know of other study that uses several zircon trace element indicators coupled with whole-rock petrogenetic analyses to assess the multi-million year evolution of a continental arc in an extensional tectonic setting (most of them deal with arcs developed in contractional regimes). Most studies based on zircon trace element determinations are devoted to particular igneous or metamorphic units, or are restricted to the use of one or two indicators of the magmatic conditions in the arc.

There is no link between the tectonic model for this area and that for the rest of the Andes, or for this type of subduction zone setting elsewhere in the world – both of these would significantly increase the impact of the current study.

A: In the last subsection of the manuscript (before the Methods section), we have included a paragraph to address this specific comment.

Some smaller comments

-

The zircon data appear to fit with the WR geochemical data, although I'm relatively sure that the same conclusions could be drawn from the WR data themselves without the need for the zircon chemistry data, apart from the oxygen fugacity data.

A: We do not dismiss the use of whole-rock (WR) geochemistry, on the contrary, we believe that by combining zircon petrochronology with whole-rock data we can reach more robust interpretations. But to related geochemical data with specific events requires precise geochronology such as U-Pb zircon dating.

Furthermore, all analytical techniques and methods have their pros and cons. In particular, some shortcomings of WR geochemistry are:

- Most WR analyses (previous studies) lacks precise radiometric ages, precluding a proper assessment of the evolution of the magmatic arc;
- The number of WR analyses is significantly less than the data point (grains) of zircon trace element determinations, making it more difficult to assess trends and avoid the effect of outliers;
- Even though the samples selected for WR geochemistry are mostly unaltered, most rocks in the early Andes of northern Chile have superimposed low degrees of hydrothermal or metamorphic alteration. This could be relevant for some parameters used in this study.

The oxygen fugacity data presented are also not necessarily in full agreement with the statements in the manuscript – for example, the on line 171, the authors indicate the the oxygen fugacity decreases during the first two magmatic episodes, however the values fluctuate between ~ -1.5 to ~ -4 , the zircon Ce anomaly doesn't change much nor does the Eu anomaly, which has recently been linked to depth of melting, so may not only be effected by oxygen fugacity (Tang et al. 2020 - doi.org/10.1130/G47745.1).

A: We agree. The oxygen fugacity calculations (ΔFMQ) do not show a clear trend, specifically for the first two periods (Late Triassic-Early Jurassic). This could be related to few data points and no useful information from previous studies. Therefore, outliers and specificities of the samples could have a significant impact on the interpretation. Nevertheless, a slight decrease in the “average” values can be observed in Figure 2. Regarding the zircon Ce/Nd and zircon Eu-anomaly, we agree that the observed trend is minor. Therefore, we have changed the sentence in line 171 to “Oxygen fugacity markers slightly decrease in this period from intermediate initial values”.

The authors suggest that the zircon trace element data is required because there is the possibility that the WR data has been altered, however the WR data are the data that most clearly show the origin of the samples and the different conditions in the source.

A: As stated above, we agree with the reviewer’s comment. The strength of this paper is our multi-proxy geochemical approach, based on both zircon petrochronology and whole-rock analyses, to determine the variable contributions of mantle-crust-slab-sediments to the magma source to better understand the early evolution of the Andean continental arc.

The WR data could be also used to constrain the Ti activity, for the Ti temperature calculations. The Ti activity is currently treated in a relatively simplistic manner – I assume the same value is used for all of the samples? Is this assumption justified? Could changes in rock type, mineralogy, water content, play a role here?

A: We plotted the zircon U-Pb age vs Ti content in ppm for each point analysis (zircon grains). We included reference temperature lines using the Ferry and Watson (2007) methodology. To do so, we considered $a[\text{SiO}_2] = 1$ and $a[\text{TiO}_2] = 0.5$ and 0.7 , according to the results obtained by Schiller and Singer (2019) for traditional I-type granitoids from continental arcs. Similar parameters were used by Chelle-Michou et al. (2014), Buret et al. (2016), Padilla et al. (2016) and Wall et al. (2018), among others. We have added information on how the reference lines were calculated in this version.

We could calculate the $a[\text{SiO}_2]$ and $a[\text{TiO}_2]$ for each sample using the WR data, as proposed by the reviewer. However, this will result in a limited data set of 30 values (number of samples with WR data) geochemistry and Ti content in zircon), or we could use the same $a[\text{SiO}_2]$ and $a[\text{TiO}_2]$ for every zircon grain in each sample. Instead, we preferred to plot every zircon grain analysis with these reference lines to have a better perspective of the general trends and the changes in “average or median” single grain crystallization temperatures. Of course, this implies a general assumption, but as we stated previously it is the way in which this information is presented in several studies using zircon trace element analysis (see Chelle-Michou et al. 2014, Buret et al. 2016, Padilla et al. 2016 and Wall et al. 2018).

The difference between the age distribution for the WR samples and those for the zircon grains could be explained by problems with the K-Ar data? These could be removed.

A: The K-Ar data are broadly consistent with the more precise U-Pb ages. Nevertheless, we interpret the relatively small differences between WR and zircon ages as caused by fast cooling (K-Ar closure temperatures) and magmatic crystallisation temperatures (U-Pb zircon ages).

Are all the zircon ages plotted in the zircon distribution diagram or just the zircons thought to date the emplacement of the intrusion? Its not clear from the figures presented here, but are there antecrystic zircon or xenocrystic zircon, or zircon with suspected Pb loss present in the datasets? Are these plotted too? Are these data also included in the trace element diagrams? Are cut offs used for zircon trace element

concentrations that could be affected by secondary alteration (see Bell et al. 2019 - doi.org/10.1016/j.chemgeo.2019.02.027)?

A: Only magmatic zircon grains representative of the main plutonic events were discussed and plotted in this manuscript. The few antecrystic and xenocrystic zircons, and all other zircon grains with “unusual” chemical characteristics were excluded.

This was carried out by using different methods and criteria:

- Textural inspection by using CL-SEM imaging; anomalous grains were excluded from the LA-ICP-MS analysis.
- Zircon grains with discordant ages were excluded; identification was done via concordia diagrams and concordia statistics, and then by weighted average age diagrams and statistics.
- Zircon grains with anomalous trace element concentrations were also excluded; this was done using the limits proposed by Grimes et al. (2015) and other references. Zircon with one or more of the following criteria were excluded: La > 6 ppm, Ce > 100 ppm, Ti > 100 ppm. These values are indicators of the presence of mineral inclusions such as apatite, monazite, titanite and/or rutile; Fe, Ca > 190 ppm, to avoid altered zircon grains.
- Finally, the following discrimination criteria was used to eliminate non magmatic or metamict zircon: the Th/U ratio values of Williams and Claesson (1987) and Rubatto and Gebauer (2000) were used to discriminate between metamorphic and magmatic zircon; the La vs Sm_N/La_N fields defined by Hoskin (2005) to discriminate between magmatic and hydrothermal zircon; and the Th+U vs La_N/Gd_N fields of Whitehouse and Kamber (2002) to identify metamict zircon.

I'm not sure why the data from Jara et al. 2021 are included in the zircon data table here? these should be included in the data table with the previously published data.

A: Agreed. We have moved these data to a separate file (Supplementary File 2).

The secondary reference data need to be included for trace element and U-Pb data.

A: We have included this information as requested by the reviewer.